# Unexpected Candidal Hyphae in Oral Mucosa Lesions—A Clinico-Pathological Study

**DOI:** 10.3390/antibiotics11101386

**Published:** 2022-10-10

**Authors:** Jeremy Edel, Osnat Grinstein-Koren, Orit Winocur-Arias, Ilana Kaplan, Anna Schnaiderman-Shapiro, Amos Buchner, Marilena Vered, Ayelet Zlotogorski-Hurvitz

**Affiliations:** 1Department of Oral Pathology, Oral Medicine and Maxillofacial Imaging, Goldschleger School of Dental Medicine, Tel-Aviv University, Tel-Aviv 6997801, Israel; 2Institute of Pathology, Rabin Medical Center, Petach Tikva 4941492, Israel; 3Institute of Pathology, Sheba Medical Center, Tel Hashomer, Ramat Gan 5262000, Israel; 4Department of Oral and Maxillofacial Surgery, Rabin Medical Center, Petach Tikva 4941492, Israel

**Keywords:** candida albicans, candidiasis, hyphae, mechanical force, dysplasia

## Abstract

Background: Oral mucosal biopsies might harbor candidal hyphae (CH) in the absence of any clinical signs or symptoms. Aim: To assess oral mucosa biopsies for the frequency of unexpected CH and characterize their clinico-pathological features. Materials and Methods: All biopsy reports (2004–2019) were searched using CH/candida/candidiasis as key words. Cases with clinical diagnosis of oral candidiasis (OC) were excluded. Demographic data, health status, smoking habits, clinical features and diagnoses were collected. Statistical analysis included the chi-square test; significance was set at *p* < 0.05. Results: Of all the biopsies, 100 (1.05%) reported microscopical evidence of CH without typical clinical signs/symptoms of OC. Fifteen cases were from healthy, non-smoking patients. CH was common on buccal mucosa (38%) and lateral tongue (23%). The tip of tongue (OR = 54.5, 95% CI 9.02–329.4, *p* < 0.001) and lateral tongue (OR = 3.83, 95% CI 2.4–6.09, *p* < 0.001) were more likely to harbor CH-positive lesions. CH-positive lesions were diagnosed as epithelial hyperplasia (55%) and exophytic reactive lesions (30%). No correlation was found between CH and the grade of epithelial dysplasia. Conclusions: Microscopic evidence of CH embedded into oral epithelium without typical signs/symptoms of OC is rare, especially in healthy, non-smokers. Since CH was occasionally found in oral sites prone to local trauma and in association with reactive lesions, in absence of host co-morbidities, the contribution of local mechanical forces to CH embedment cannot be ruled out.

## 1. Introduction

In the oral cavity, candida, commonly of the albicans type, is the most frequent fungal microorganism [1], normally found in up to 60% of healthy individuals [2]. Morphologically, the vegetative blastopores (yeast cells) represent the commensal phase; in favorable environments, these transform into hyphae, which represent the pathogenic phase of the fungus [3,4]. Hyphae are less phagocytized by the immune cells, enabling the fungus to escape from the macrophages and neutrophils, and, together with their penetration, subsequent destruction of the host tissues occurs [5].

A recent review on the cell wall of candida, focused on mutations in gene-encoding cell-wall-associated proteins with a role in hyphal morphogenesis, showed that mutations in specific proteins affect hyphae generation, with resulting attenuated fungal virulence [6]. In addition to the mutational status of the fungus, a review of the complex myriad of candida–host interactions has emphasized that factors such as the specific composition of the organ microbiota (e.g., oral, gut, vaginal), and especially the host antifungal immunity, will build up the individual susceptibility to infection [7]. Factors that may influence the pathological transformation of candida in the oral cavity also include malnutrition, metabolic disease, salivary gland hypofunction, local trauma and presence of dental prostheses/devices [8,9].

The lesions caused by candida in the oral cavity, oral candidiasis (OC), have been categorized into several defined forms, depending on the clinical appearance, underlying health status and local factors: pseudomembranous, acute erythematous/atrophic, chronic erythematous/atrophic, median rhomboid glossitis and chronic hyperplastic candidiasis [2,10]. Biopsies taken from oral mucosa affected by candidiasis may vary slightly, depending on the clinical form, but essentially share the same microscopic findings of para-keratinized (less frequently, non-keratinized) epithelial surface, varying degrees of epithelial hyperplasia, collections of neutrophils within the keratin and upper spinous layer, stromal chronic inflammatory cells and candidal hyphae (CH) embedded in the keratin layer, only down to the uppermost cells of the spinous layer [11]. Nonetheless, CH have also been reported in oral mucosa lesions submitted for microscopic examination for different reasons, without any clinical suspicion of OC [1,12]. Similar observations were also found extra-orally, as recently reported in adolescent women with clinically asymptomatic vaginal candidal colonization [13].

The aim of the present study was to characterize the clinico-pathological features of oral soft tissue lesions with clinically asymptomatic and unanticipated microscopic presence of CH.

## 2. Results

### 2.1. Demographics and General Health-Related Findings

Out of a total of 9525 oral soft tissue biopsies retrieved, 100 (1.05%) reports with unexpected microscopic CH were found.

The age range of patients in the study group was 16–92 years (mean age 58.3 ± 16.7 years); the mean age of male patients (62.6 ± 17.5 years) was higher than that of female patients (52.8 ± 16.9 years) (*p* = 0.025). The distribution of patients across decades is shown in Figure 1 and emphasizes the peak number of patients in the seventh and eighth decades. A complete documentation of gender was found in 79 patients, consisting of 40 females and 39 males, with an almost 1:1 ratio.

Regarding the systemic health status and smoking (Table 1), 15 patients had a history of oral squamous cell carcinoma (OSCC), 51 had systemic health conditions, and in 51, medication intake was recorded. In all, 23 (29.1%) reported past or present tobacco smoking, of whom 15 were medically compromised, and 8 were healthy patients. Fifteen (19% of those with available data, 15% of the study group) patients were healthy and non-smokers.

### 2.2. Lesion-Related Clinical Features

Clinically, the information on the lesion’s color was recorded in only 53 cases, of which 23 (43.4%) were reported to be white, 14 (26.4%) red, 8 (15.1%) white and red, and 8 (15.1%) were the color of the surrounding mucosa. In 33 of the reports, lesions were described to be exophytic; no information was available for the remaining cases.

The most frequently involved site was the buccal mucosa 38 (38%), followed by the tongue, 37 (37%), of which 23 (62.2%) cases involved the lateral aspect, 9 (24.3%) dorsal, 3 (8.1%) ventral, and 2 (5.4%) involved the tip. The remaining lesions involved alveolar ridge/gingiva—10 (10%); palate—7 (7%); lower/upper labial mucosa and floor of mouth—1 each (1%); in 3 (3%) cases, lesions involved two or more adjacent mucosal surfaces; 2 (2%) cases lacked specified location (Table 2).

Among CH-free oral soft tissue biopsies (N = 9425), there was a different distribution of locations, with mucosa of the alveolar ridge or gingiva being the most frequently involved (29.8%), followed by the buccal mucosa (26.3%), tongue (16.7%, all aspects), lower labial mucosa (13.1%), palate (9.5%) and floor of mouth (3.8%). Upper labial mucosa and commissural area showed negligible numbers of lesions.

CH-positive lesions had prevalence in the tip of tongue (OR = 62.7, 95% CI 9.02–329.4, *p* < 0.001), lateral aspect of tongue (OR = 4.56, 95% CI 2.4–6.09, *p* < 0.001) and buccal mucosa (OR = 1.66, 95% CI 1.11–2.49, *p* = 0.013). They had a reduced chance of being found in the lower labial mucosa and alveolar ridge and gingiva (OR = 0.063, 95% CI 0.007–0.39, *p* < 0.001 and OR = 0.26, 95% CI 0.11–0.42, *p* < 0.001, respectively).

### 2.3. Lesion-Related Microscopic Diagnoses

In total, CH were found in association with microscopic diagnoses of epithelial hyperplasia in 55 (55%) cases, of which 12 presented dysplasia (12% of all cases, 21.2% of cases with epithelial hyperplasia); 30 (30%) were reactive exophytic lesions with fibrous/fibro-epithelial hyperplasia, 11 (11%) lichen planus or lichenoid lesions, 3 (3%) epithelial atrophy/ulcer and 1 (1%) psoriasiform mucositis (geographic tongue). Microscopic diagnoses significantly correlated with the clinical diagnoses (*p* = 0.001, *p* = 0.00003 with Bonferroni correction). Figure 2 shows a case of clinically suspected leukoplakia of the buccal mucosa, which microscopically showed epithelial hyperplasia, hyperkeratosis on hematoxylin and eosin (H&E)-stained slide (Figure 2A) with superimposed candidiasis, where the fungal hyphae could be seen on the H&E-stained slide (Figure 2B, yellow arrow) and were further highlighted by periodic acid Schiff (PAS) stain (Figure 2C, blue arrows). An additional case with a clinical diagnosis of irritation fibroma on the buccal mucosa, which microscopically exhibited fibro-epithelial hyperplasia, increased vascularity but almost no inflammation, is shown (Figure 2D, H&E staining). CH were observed on the H&E-stained slide (Figure 2E, yellow arrows) and further highlighted by PAS stain (Figure 2F, blue arrows).

The distribution of the microscopic diagnoses according to various oral sites is illustrated in Figure 3. Epithelial hyperplasia/hyperkeratosis (without dysplasia) with CH constituted 100% of lesions of the floor of mouth and ventral tongue, 72% of the palate, 48% lateral tongue, 34% buccal mucosa and less so of the lesions of dorsal tongue and alveolar mucosa/gingivae. Exophytic reactive lesions (fibro-epithelial hyperplasia) with CH constituted 100% of the lesions at the tip of tongue and lower lip mucosa, 50% of the alveolar ridge mucosa or gingivae, 44% of the dorsal tongue, 32% of the buccal mucosa and less so at the lateral aspect of the tongue (17%) and hard palate (14%). The probability of CH at the tongue tip was 98.4%, on the lateral tongue—82%, buccal mucosa—62.4%, and less so on the alveolar ridge mucosa/gingiva—20.6%, and least on the lower lip mucosa—5.9% (Table 2). The mean age of patients did not differ as a factor of location of microscopic diagnoses (*p* > 0.05). No significant correlations were found between the location of lesions and microscopic diagnoses (*p* > 0.05).

In 12 cases, dysplastic changes were found, which were located as follows: 6 on lateral tongue (3 mild, 2 mild-to-moderate, 1 severe), 1 dorsal tongue (mild) and 5 buccal mucosa (2 mild, 3 mild-to-moderate). Combined with the medical information, six (50%) of these cases had a history of OSCC.

Regarding the medical history of the patients, among healthy, non-smokers (N = 15), CH were found in association with epithelial hyperplasia in 12 (80%) cases, of which 1 (6.6%) was with dysplasia; and 3 (20%) were exophytic reactive lesions. Among all non-smokers (N = 58), CH were found in 37 (63.8%) of the lesions with epithelial hyperplasia, of which 10 (17.24%) were with dysplasia; and 16 (27.6%) were exophytic reactive lesions. Of the exophytic reactive lesions, the lateral tongue (N = 6) and buccal mucosa (N = 4) were the most common sites involved, followed by the tip of tongue (N = 2), alveolar mucosa (N = 2), labial mucosa and hard palate (N = 1, each).

Based on the present results, Figure 4 summarizes the various interactions between the host- and candida-related factors and the clinical outcomes.

## 3. Discussion

OC is a defined, recognized, clinico-pathological group of conditions. The aim of the present study was to identify and characterize cases, which harbor embedded CH but are devoid of clinical features suggestive of OC. The frequency of these lesions within a large series of 9525 oral mucosal biopsies, over a 16-year period of biopsy service, was found to be low, only 1.05%. These cases were mainly associated with epithelial hyperplasia/hyperkeratosis, usually without dysplasia, or with reactive exophytic fibro-epithelial hyperplasia, with a propensity for the tongue and buccal mucosae. In addition, we found an unanticipated sub-group of 15 patients who were healthy and non-smokers, in contrast to the known predisposition of CH to be associated with a setting of compromised patients and/or tobacco smoking habits [2]. The possible clinical and biological causes for this interesting finding are discussed.

Although OC is a relatively common condition, the current low frequency of lesions showing CH is probably due to the fact that, in most cases, the diagnosis of OC is clinically based, without the necessity for further histopathological evaluation [2]. Age-wise, the distribution of the CH-positive cases per decades of life showed peak frequency in the seventh and eighth decade of life, which is in line with the reported increased frequency of OC with age [14]. In addition, the definite combability between the microscopic and clinical diagnoses, which did not include OC, further sharpens the randomicity of the tissue-embedded CH. In a previous study, the CH frequency in oral mucosa biopsies routinely submitted for microscopic examination was higher—4.7% [12]. A possible explanation for this difference lies in the research methods; while in that study, periodic acid-Schiff (PAS) staining was performed for all the examined sections, in the present study, PAS was performed only for H&E-stained sections that were suspected to harbor CH. In addition, excluding all cases with clinical diagnosis of OC or antifungal pre-treatment also contributed to the present low frequency of CH.

The transition of candida from its commensal form as a yeast into a pathogenic state, morphologically, which is represented by hyphae and clinically manifested in the oral mucosa as OC, is usually an outcome of various systemic and local predisposing factors [6]. Our study group included only cases with no clinical suspicion of OC. Analysis of their medical data revealed that at least some of the patients had localized or systemic health problems or smoking habits that could promote the pathologic colonization of candida [1,2,15]; yet, these were apparently not sufficient for full-blown clinically recognized OC. More interestingly, we identified a sub-group of 15 CH-positive patients who were completely healthy and non-smokers. Previous studies have also reported hypha structures in smears taken from healthy subjects [1,16], thus suggesting that their presence was not a definite sign of candidal infection [1]. The assumption that not every case with epithelium-embedded CH inevitably results in significant spread and infection has been previously suggested by Wächtler et al. [17]. Their study showed that hyphae formation was fundamental for epithelial invasion but that epithelial invasion itself did not damage the host cells. An entire transcriptional program associated with the morphological transition and appropriate niches of fungus-host-microbiota are required for the ultimate spreading and consequently destructive effect on the host cells [7,17,18]. However, the fact that CH were presently found in defined lesions and not on normal mucosa, especially in those healthy/non-smokers, may raise the possibility that at least some of the patients might have carried systemic or local latent or undiagnosed predisposing factor/s, which would potentially promote clinical OC, given the appropriate conditions.

Similar to Barrett et al. [12], we also found that lesions that harbored embedded CH were most frequently located on the tongue (38%) and buccal mucosa (37%), which are often the sites exposed to mechanical trauma. This site distribution significantly differed from that found in CH-negative oral mucosa lesions, suggesting a non-randomized distribution of the fungal-associated lesions. We also showed that CH were mostly associated with epithelial hyperplasia (55%) and exophytic reactive lesions (30%). Taking into consideration the location, types of CH-related lesions and the sub-group of healthy, non-smokers, some relationship between CH and local mechanical forces can be suggested.

Previous studies have proposed a link between local mechanical forces and the tendency for candidal infection, as in the case of denture stomatitis [19,20], although this concept had been contradicted by others [21,22]. Nevertheless, a link between mechanical force and candidal invasion could be deduced by integrating the findings from the two—oral mucosae and candida. Oral-mucosa-wise, Nakamura et al. have found that the application of mechanical force, such as dentures-related occlusal pressure, induced intracellular stress, inflammation and diminished intercellular adhesion [23].

Candida-wise, two parallel but complimentary mechanisms of invasion into the epithelial cells during the pathogenic stage were suggested: candida-induced endocytosis, which is a clathrin-mediated and actin-dependent process, and an active penetration into the host cells, probably achieved by the physical forces induced by the elongating fungal hyphae combined with secreted hydrolytic enzymes. The latter mechanism facilitates the fungus to penetrate through both the cell membrane and intercellular gaps of the epithelial tissue [24]. Collectively, diminished epithelial intercellular adhesions caused, for example, by external mechanical force may theoretically ease candidal invasion. It is well established that in order to survive, the fungus must rapidly respond to local environmental changes, such as pH, nutrients, osmolarity and oxidative stress, by altering its transcriptional profile [18]. Nonetheless, evidence of the influence of external mechanical force on the fungus response is still lacking. The initial evidence in this context comes from the study by Alsteens et al. who, by using a single-molecule atomic force microscopy, have shown that mechanical stimuli could trigger adhesion nanodomains in candidal cells [25], inferring that force-induced clustering of adhesins may be a mechanism for activating candidal cell adhesion needed for further interaction with oral epithelial cells. In this line, it would be interesting to note that the alveolar mucosae and gingivae were one of the less likely locations for the presence of embedded CH in spite of them being exposed to continuing mechanical trauma. A feasible explanation would be the nature of the oral mucosa at these sites, where the epithelium is very condensed, with no inter- or intra-cellular edema, and it is tightly attached to the underlying dense connective tissue. Therefore, the seemingly reduced intercellular gaps and surface cavitations/depressions could impede the morphogenic hyphal transformation and embedment into the alveolar/gingival epithelium. Alternatively, it is possible that as a result of the keratinized mucosa at these sites being a proper setting for the keratinophilic candida [11] to raise a significant infection with clinical manifestation, it was the clinical disease that led to exclusion of these cases from the study group. Figure 3 summarizes the various interactions between the host- and candida-related factors and the clinical outcomes.

In the present study, dysplastic changes of variable degrees (six mild, six mild–moderate, two severe) were noticed in 12% of all CH-positive cases, almost a quarter of the lesions with epithelial hyperplasia/hyperkeratosis in this group. In six cases, there was a history of OSCC, raising the possibility that local host factors, such as hyposalivation following the oncologic treatment [26], promoted the fungal colonization on already altered mucosa [27]. Clearly, our results showed no correlation between the presence of CH and the severity of dysplasia. Similar results were reported by Singh et al. who investigated 50 individuals with lesions associated with areca nut and tobacco usage [28]. On the contrary, other authors have observed a significant association between candidiasis and advanced degree of dysplastic changes [12,29,30,31] and even with worsening the degree of dysplasia over time [32]. This was attributed to the fungal capacity to produce N-nitrosobenzylmethylamine, a potent carcinogen [33,34]. Differences in the research methods may explain the dissimilarity between studies. However, the possible role of candida in oral mucosa transformation still remains to be consolidated.

Although our study perused a large number of biopsies, only a relatively small number of oral mucosa lesions with embedded CH were found—a fact, which probably represents the actual prevalence of our findings. Nevertheless, these cases were collected only from a central area in Israel, and it can be assumed that demographic populations that differ in parameters, such as smoking habits, alcohol consumption, nutrition constituents, could have provided different rates of clinically asymptomatic, microscopically embedded CH lesions. The importance of our findings lies in raising awareness of the existence of this sub-group of patients among the community of relevant clinicians and researchers, as this should entail an in-depth clinical and molecular research aimed to elucidate the etiology and pathogenesis of this condition. So far, it can be recommended that, once the histopathological report is positive for embedded CH, post-operative follow-up should be commenced to monitor the potential progression to overt OC as well as evolvement of any latent medical disorder. The elimination of sources of local mechanical trauma should always be encouraged.

In conclusion, there are a few cases with microscopic evidence of CH embedded into oral epithelium without typical signs/symptoms of OC, including healthy/non-smoking individuals. This finding could be referred to as “hidden” OC. It can be assumed that the oral mucosae harbor focal niches, where local mechanical trauma can facilitate the morphological transformation of candida into a potentially pathogenic state, leading to focal occult OC. More concrete evidence of the mechanism driving the mechanical-force-related, pathological transformation of candida in the oral mucosae is needed.

## 4. Materials and Methods

The study protocol was reviewed and approved by the Ethics Committee of Tel Aviv University and IRB of Sheba Medical Center (SMC-19-6666) and is in compliance with the Helsinki Declaration.

For this retrospective study, all oral soft tissue biopsy reports accessed between 2004 and 2019 were searched, using the key words candida, candidal hyphae (CH) or candidiasis in the microscopic report. All cases that fulfilled these criteria were included in the study group. For this group, data of the patients’ demographics (gender, age), clinical details regarding the health status (systemic health conditions, permanent medications, history of oral cancer and smoking habits), lesions’ clinical features of site of involvement, color (white, red, mixed white and red, same color as the normal adjacent oral mucosa) and growth pattern (exophytic or not) and the submitted clinical diagnosis were retrieved from the records. All other soft tissue biopsies devoid of microscopic evidence of CH referred to us within the same framework of time were considered as the control group. For this group, only the location of the lesions was collected.

In general, all the oral lesions were biopsied by scalpel, with patients under local anesthesia. Those defined, exophytic, pedunculated or broad-based lesions were usually removed per excisional biopsy. Flat, extensive and usually less well-defined lesions were sampled as an incisional biopsy procedure. The removed tissues were fixed in 10% formalin solution for 24 h and then processed by an automatic benchtop tissue processor (Leica TP1020, Leica Biosystems, Deer Park, IL, USA). All tissues were then routinely stained with hematoxylin and eosin (H&E). Four-micron-wide sections were prepared and then routinely stained with hematoxylin and eosin (H&E).

With regard to histopathological diagnoses, there were six classes of lesions: epithelial hyperplasia/hyperkeratosis (clinical white lesions of leukoplakia), epithelial hyperplasia/hyperkeratosis and dysplasia (clinical mixed white and red lesions—erythroleukoplakia), epithelial atrophy/ulcer (clinical erythroplakia or suspected for squamous cell carcinoma), lichen planus/lichenoid lesions, psoriasiform mucositis (clinical geographic tongue) and reactive exophytic lesions/fibro-epithelial hyperplasia (i.e., irritation fibroma/fibrous epulis with/without inflammation).

Initially, all cases in which CH were microscopically observed were collected. In the presence of an appropriate clinical and histological appearance, a diagnosis of OC was submitted. Cases with a microscopic diagnosis of OC that corresponded to the clinical suspected diagnosis and those cases for which we had information about treatment with anti-fungal agents before biopsy were excluded. The study group included cases of any other defined oral mucosal lesion in which CH was identified within the keratin layer/superficial epithelial layers (on H&E-stained sections or PAS stain). If there was a corresponding inflammatory reaction, a diagnosis of a specific lesion with superimposed candidiasis was given, and in the absence of an inflammatory reaction, a note was added to the diagnosis describing the presence of CH. Two oral pathologists (IK, MV) separately reviewed the cases, and those that raised disagreement were decided by mutual accordance.

Statistical analysis included the investigation of associations between the location of the study group (oral lesions with CH) and the control group (oral lesions without CH) and was performed by crosstab with Fisher’s exact test with Bonferroni post hoc correction, followed by calculation of the odds ratio (OR) and probability. The mean age of patients was analyzed as a factor of gender, location of lesions and microscopic diagnoses. For these analyses, we used ANOVA with Bonferroni post hoc corrections. Correlations between microscopic and clinical diagnoses and microscopic diagnoses and lesion locations were analyzed by crosstabs and Pearson chi-square test. The distribution of normality was analyzed by the Kolmogorov–Smirnov test, with a resulting *p* = 0.14. All statistical workup was performed using the SPSS software, version 27 (IBM, Chicago, IL, USA), and statistical significance was set at *p* < 0.05.

## Figures and Tables

**Figure 1 antibiotics-11-01386-f001:**
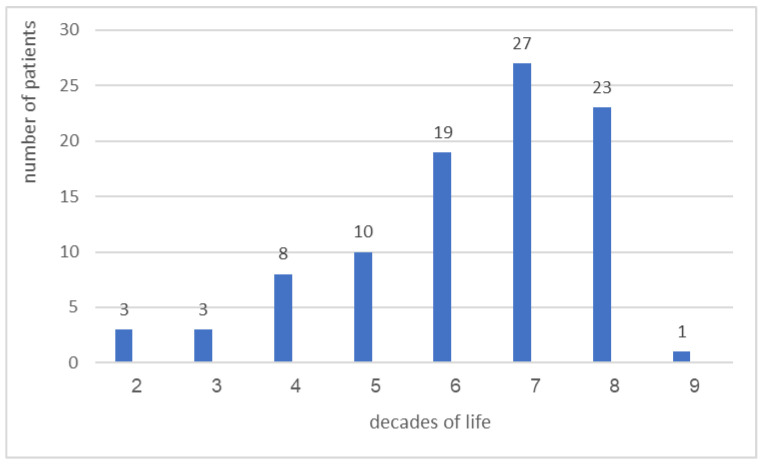
Distribution of patients with lesions containing embedded CH according to decades of life.

**Figure 2 antibiotics-11-01386-f002:**
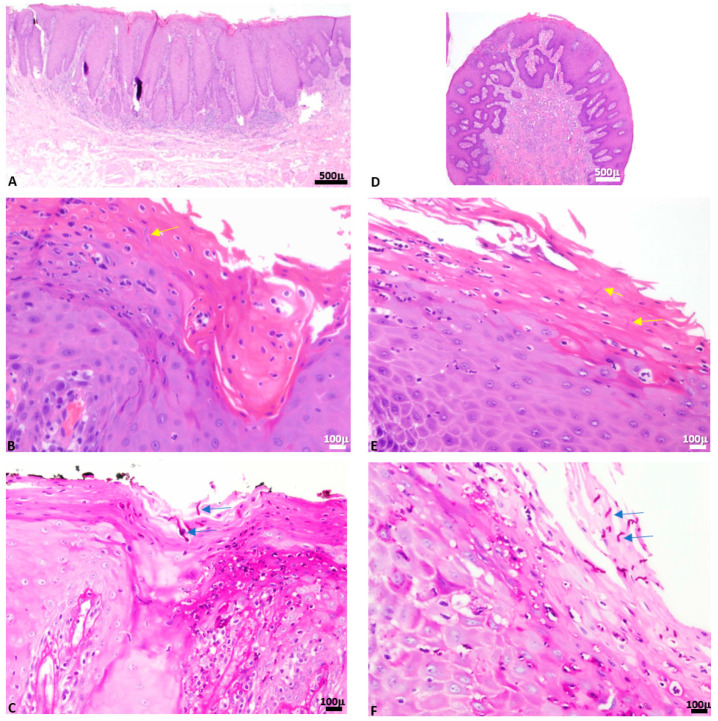
A case of clinically suspected leukoplakia of the buccal mucosa, which microscopically showed epithelial hyperplasia with hyperkeratosis and superimposed candidiasis (**A**–**C**), and a lesion that had a clinical diagnosis of an exophytic reactive lesion, consistent with irritation fibroma of the buccal mucosa with CH (**D**–**F**).

**Figure 3 antibiotics-11-01386-f003:**
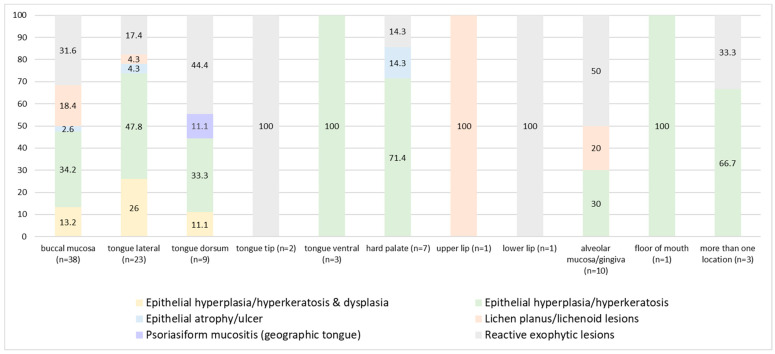
Distribution (%) of microscopic diagnoses in the study group (N = 98) as a factor of location in the oral cavity.

**Figure 4 antibiotics-11-01386-f004:**
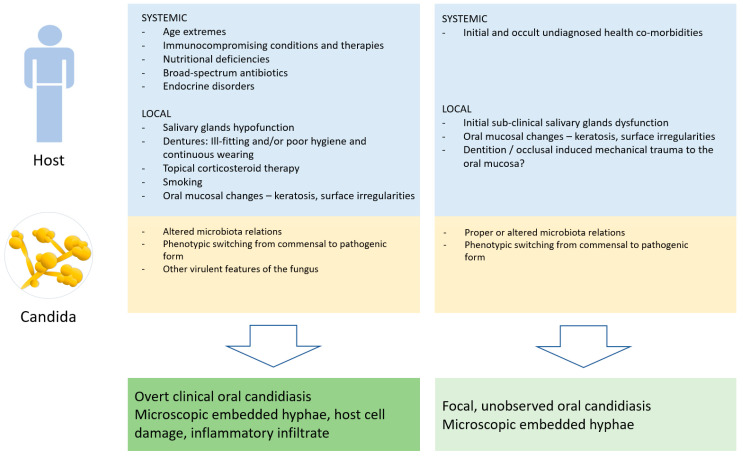
Interactions between host- and candida-related factors and the clinical outcomes.

**Table 1 antibiotics-11-01386-t001:** Distribution of lesions with candidal hyphae.

			Smoking	History of OSCC	Systemic Conditions	Systemic Medication
**Medically compromised (N = 56)**	Hyperplasia, hyperkeratosis	With dysplasia N = 10	1 (10%)	6 (60%)	9 (90%)	8 (80%)
		Without dysplasia N = 24	8 (33.3%)	6 (25%)	21 (87.5%)	21 (87.5%)
		Total N = 34	9 (26.5%)	12 (35.3%)	30 (88.2%)	29 (85.3%)
	Irritation fibroma and other reactive lesions N = 17		4 (23.5%)	3 (17.6%)	16 (94.1%)	15 (88.2%)
	Lichen planus/lichenoid reaction N = 5		2 (40%)	0	5 (100%)	5 (100%)
**Healthy (N = 23)**	Hyperplasia, hyperkeratosis	With dysplasia N = 1	0	-	-	-
		Without dysplasia N = 17	6 (35.3%)	-	-	-
		Total N = 18	6 (33.3%)	-	-	-
	Irritation fibroma and other reactive lesions N = 5		2 (40%)	-	-	-
	Lichen planus /lichenoid reaction N = 0		0	-	-	-

N—refers to the number of cases in which information was available; OSCC—oral squamous cell carcinoma.

**Table 2 antibiotics-11-01386-t002:** Comparison of lesion locations between the study group and the control group (Fisher’s exact test).

Location		Candidiasis		*p* Value *	OR/Probability
		Yes Study Group N = 100	No Control Group N = 9425		
**Tongue lateral ****	Yes	**23 (23%)**	**565 (6%)**	**<0.001**	**4.56/82%**
	No	77 (77%)	8860 (94%)		
**Tongue dorsal**	Yes	9 (9%)	772 (8.2%)	0.64	
	No	91 (91%)	8653 (91.8%)		
**Tongue ventral**	Yes	3 (3%)	184 (1.95%)	0.63	
	No	97 (97%)	9241 (98.05%)		
**Tongue tip ****	Yes	**2 (2%)**	**3 (0.02%)**	**<0.001**	**62.7/98.43%**
	No	98 (98%)	9422 (99.98%)		
**Buccal mucosa ****	Yes	38 (38%)	2474 (26.3%)	**0.013**	**1.66/62.4%**
	No	62 (62%)	6951 (73.7%)		
**Palate**	Yes	7 (7%)	893 (9.5%)	0.2	
	No	93 (93%)	8532 (90.5%)		
**Upper labial mucosa**	Yes	1 (1%)	64 (0.7%)	0.05	
	No	99 (99%)	9361 (99.3%)		
**Floor of mouth**	Yes	1 (1%)	360 (3.8%)	0.94	
	No	99 (99%)	9065 (96.2%)		
**Lower labial mucosa *****	Yes	**1 (1%)**	**1283 (13.1%)**	**<0.001**	**0.063/5.9%**
	No	99 (99%)	8142 (86.4%)		
**Alveolar ridge/gingiva *****	Yes	**10 (10%)**	**2807 (29.8%)**	**<0.001**	**0.26/20.6%**
	No	90 (90%)	6618 (70.2%)		

* Original *p* value of *p* < 0.001 was maintained after Bonferroni correction, and the original *p* value of 0.013 yielded a value of 0.003 after Bonferroni correction; ** Higher frequency of involvement of this site in the study group compared to the control; *** Lower frequency of involvement of this site in the study group compared to the control; OR—odds ratio.

## Data Availability

The data presented in this study are available on request from the corresponding author.

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
