# Peer review of "Unexpected Candidal Hyphae in Oral Mucosa Lesions—A Clinico-Pathological Study"

_antibiotics, 2022, doi:10.3390/antibiotics11101386_

Round 1

Reviewer 1 Report

Dera Authors,

Your study „Unexpected Candidal Hyphae in Oral Mucosa Lesions - A Clinico-Pathological Study” is quite interesting but now it is too generally and in this form I am not able to accept it to the publication.

Major changes:

1.       Please improve introduction section, in this form it is too general.

2.       Material and method section should be between introduction and results.

3.       What kind of statistical distribution was observed? Please improve statistical analysis, you used only chi square test. Correlation between age, location of lesion clinical and microscopic view of lesion, etc. may improve your study.

4.       Figure 2 – in second column 26%+48%+4%+4%17%=99% it is not equal 100%, please explain, correct?

5.       Line 281: In the present study dysplastic changes of variable degrees (6 mild, 6 mild-moderate, 2 severe) – 6 – mild and 6-mild-(2)-moderate please explain and correct.

6.       Add conclusions

7.       Add study limitations

8.       Only 6 of 29 references are from last 5 years, please update it.

Minor changes:

1.       Figure 1 – please add scale bar to each photography.

2.       Table 2 – please add information od statistical test in caption of table.

Best regards,

Reviewer.

Author Response

We thank the reviewer for the constructive comments. Our point-by-point responses follow. We hope that the improved version of the manuscript is now appropriate for publication.

Sincerely, The authors

Major changes:

  1. Please improve introduction section, in this form it is too general.

Response: We thank the Reviewer for this comment. The Introduction has been revised and new references were added, including more specific data that better lead to the aim of the present study (lines 39-59, 71-73).

  1. Material and method section should be between introduction and results.

Response: The manuscript was edited according to the journal's instructions and supplied word template, according to which the Material and Methos section comes after Discussion (please see - https://www.mdpi.com/journal/antibiotics/instructions)

  1. What kind of statistical distribution was observed? Please improve statistical analysis, you used only chi square test. Correlation between age, location of lesion clinical and microscopic view of lesion, etc. may improve your study.

Response: The entire paragraph on the statistical analysis has been revised, with detailed and specific reference to tests that were used (lines 554-561)

  1. Figure 2 – in second column 26%+48%+4%+4%17%=99% it is not equal 100%, please explain, correct?

Response: Initially we rounded the percent to whole numbers, however following this comment we introduced a revised figure with percent showing decimal numbers, where all sum up to 100%

  1. Line 281: In the present study dysplastic changes of variable degrees (6 mild, 6 mild-moderate, 2 severe) – 6 – mild and 6-mild-(2)-moderate please explain and correct.

Response: degrees of dysplasia are not always uniform along the entire length of the lining epithelium and sometimes there is a gradient of severity or, alternatively, there is a general low degree of dysplasia with foci of more aggravated changes. In these cases, the histopathological diagnosis can reflect a range of severity of dysplasia, like mild-to-moderate, and this was the case with 6 of the cases diagnosed with dysplasia in our study   

  1. Add conclusions

Response: Conclusions were highlighted (lines 447-453)

"In conclusion, there are a few cases with microscopic evidence of CH embedded into oral epithelium without typical signs/symptoms of OC, including healthy/non-smokers individuals. This finding could be referred to as "hidden" OC. It can be assumed that the oral mucosa harbor focal niches, where local mechanical trauma can facilitate the morphological transformation of candida into a potentially pathogenic state, leading to focal occult OC. More concrete evidence for the mechanism driving the mechanical force-related, pathological transformation of candida in the oral mucosae, is needed."

  1. Add study limitations

Response: Limitations were added (lines 433-446)

" Although our study perused a large number of biopsies, only a relatively small number of oral mucosa lesions with embedded CH was found, a fact that probably represents the actual prevalence of our findings. Nevertheless, these cases were collected only from a central area in Israel and it can be assumed that demographic populations that differ in parameters like smoking habits, alcohol consumption, nutrition constituents, could have provided different rates of clinically asymptomatic, microscopically embedded CH lesions. The importance of our findings lies in raising awareness on the existence of this subgroup of patients among the community of relevant clinicians and researchers, as this should entail an in-depth clinical and molecular research aimed to elucidate the etiology and pathogenesis of this condition. So far, it can be recommended that once the histopathological report is positive for embedded CH, post-operative follow-up should be commenced to monitor potential progression to overt OC as well as evolvement of any la-tent medical disorder. The elimination of sources for local mechanical trauma should be always encouraged."

  1. Only 6 of 29 references are from last 5 years, please update it.

Response: We thank the reviewer for this note. In general, clinically overt oral candidiasis has been extensively investigated but we could not find any recent reports in the literature on those cases of "hidden"/subclinical oral candidiasis that we describe in the present manuscript. The basic knowledge on oral candidiasis is also "classical" literature from a few years ago. Following the Reviewer's comment, we added a number of recent publications, as marked in the reference list (#5-7, 9, 13, 14).

Minor changes:

  1. Figure 1 – please add scale bar to each photography.

Response: Scale bars were added and the updated figure replaced the original one

  1. Table 2 – please add information od statistical test in caption of table.

Response: Statistical test was added in caption of Table 2

Reviewer 2 Report

Good day,

Dear Authors,

Thank You for pleasure to read Your work.

I have several comments to improve Your work.

Sincerely, Reviewer

Title.

I am not sure that in English the word ‘ Clinico-Pathological’ is present.. please, check it according to MeSH.

Abstract

Please, write it with structure ‘Actuality-Aim-Materials and Methods- Results-Conclusion’.

Check Key words according to MeSH.

Introduction

It is too small. Please, add information about the clinical importance of ‘candidal hyphae’.

‘The aim of the present study was to survey biopsied oral soft tissue lesions for the 56 cases with unanticipated microscopic presence of CH and characterize their frequency 57 and clinico-pathological features.’ Here I see at least 3 aims or aim and two tasks. Please, chose one.

Material and Methods

First, please, use the article structure according to STROBE requirements.

In this form it is rather hard to read Your manuscript.

Please, write probability for table 1 (in text or in the additional column).

Line 80 ‘The most frequently involved site was the buccal mucosa 38 (38%) followed by the 80 tongue, 37 (37%) cases, of which 23 (62.2%) involved the lateral aspect, 9 (24.3%) dorsal, 3 81 (8.1%) ventral, and 2 (5.4%) involved the tip.’ Is it right for 38 patients to be 38%? Please, check

Section 2.3

Figure 1. Legend is too big; the explanation must be in the text. For legend use names but not full description.

For figure 2, please, write probabilities (in text or mark them with * in case of significancy).

Discussion

The figure 3 is not obligate or add it in the appropriate section if You talk about Your own results.

Materials and methods

Ethical approval. Your research was from 2004 year, but Board approval was dated Sept. 2013. How could You explain this difference?

Please, describe the zone and methods of biopsy/material taking.

Statistics. It is too small description, please, write this section with details. Also, you have many comparisons that require a correction. Please, write results according to Bonferroni of FDR correction.

Section ‘Conclusion’ is missed. Please, write it.

Author Response

We thank the reviewer for the constructive comments. Our point-by-point responses follow. We hope that the improved version of the manuscript is now appropriate for publication.

Sincerely, The authors

  1. Title.

I am not sure that in English the word ‘Clinico-Pathological’ is present.. please, check it according to MeSH.

Response: the term "Clinico-pathological" is mentioned in MeSH at https://www.ncbi.nlm.nih.gov/mesh/?term=clinico+pathological:

Clinical Conference [Publication Type]: Work that consists of a conference of physicians on their observations of a patient at the bedside, regarding the physical state, laboratory and other diagnostic findings, clinical manifestations, results of current therapy, etc. A clinical conference usually ends with a confirmation or correction of clinical findings by a pathological diagnosis performed by a pathologist. "Clinical conference" is often referred to as a "clinico-pathological conference." Year introduced: 2008(1991)

  1. Abstract                                                                                                      - Please, write it with structure ‘Actuality-Aim-Materials and Methods- Results-Conclusion’.

Response: The Abstract and the rest of the manuscript were written according to the instruction of the journal found at https://www.mdpi.com/journal/antibiotics/instructions:

"Abstract: The abstract should be a total of about 200 words maximum. The abstract should be a single paragraph and should follow the style of structured abstracts, but without headings: 1) Background: Place the question addressed in a broad context and highlight the purpose of the study; 2) Methods: Describe briefly the main methods or treatments applied. Include any relevant preregistration numbers, and species and strains of any animals used. 3) Results: Summarize the article's main findings; and 4) Conclusion: Indicate the main conclusions or interpretations. The abstract should be an objective representation of the article: it must not contain results which are not presented and substantiated in the main text and should not exaggerate the main conclusions".

3. Check Key words according to MeSH.

Response: all key words have been checked in the MeSH, and the results are as following:

  1. Candidiasis - "candidiasis"[MeSH Terms] OR candidiasis[Text Word] - 13
  2. Candida albicans - albicans[All Fields] AND ("candida"[MeSH Terms] OR candida[Text Word]) – 104 or "candida albicans"[MeSH Terms] OR candida albicans[Text Word] - 102
  3. Hyphae – Hyphae Microscopic threadlike filaments in FUNGI that are filled with a layer of protoplasm. Collectively, the hyphaemake up the MYCELIUM. Year introduced: 2002 (available at: https://www.ncbi.nlm.nih.gov/mesh/?term=hyphae&cmd=DetailsSearch)
  1. Mechanical force - mechanical[All Fields] AND force[All Fields] (12)
  2. Dysplasia - dysplasia[All Fields] (480)

4. Introduction

- It is too small. Please, add information about the clinical importance of ‘candidal hyphae’.

Response: We thank the Reviewer for this comment. The Introduction has been revised and new references were added, including more specific data that better lead to the aim of the present study (lines 39-59, 71-73).

-‘The aim of the present study was to survey biopsied oral soft tissue lesions for the cases with unanticipated microscopic presence of CH and characterize their frequency and clinico-pathological features.’ Here I see at least 3 aims or aim and two tasks. Please, chose one.

Response: the aim was changed as suggested: "The aim of the present study was to characterize the clinico-pathological features of oral soft tissue lesions with unanticipated microscopic presence of CH" (lines 74-76)

5. Material and Methods

- First, please, use the article structure according to STROBE requirements.

In this form it is rather hard to read Your manuscript.

Response: In addition to the fact that the manuscript was prepared according to the Journal's instructions (https://www.mdpi.com/journal/antibiotics/instructions), we have carefully read all STROBE requirements and checked that the manuscript addressed all sections that were relevant to the present study

- Please, write probability for table 1 (in text or in the additional column).

Response: probabilities were added in Table 1, as suggested

- Line 80 ‘The most frequently involved site was the buccal mucosa 38 (38%) followed by the tongue, 37 (37%) cases, of which 23 (62.2%) involved the lateral aspect, 9 (24.3%) dorsal, 3 (8.1%) ventral, and 2 (5.4%) involved the tip.’ Is it right for 38 patients to be 38%? Please, check

Response: In respect to buccal mucosa, 38 patients out of 100 are 38%. In respect to tongue location – 37 patients out of 100 are 37%, etc.

6. Section 2.3

- Figure 1. Legend is too big; the explanation must be in the text. For legend use names but not full description.

Response: The legend has been shortened and most of it has been introduced into the text, as suggested (lines 170-179)

 - For figure 2, please, write probabilities (in text or mark them with * in case of significancy).

Response: Probabilities were noted in text according to suggestion (now Figure 3, lines 216-247)

7. Discussion

- The figure 3 is not obligate or add it in the appropriate section if You talk about Your own results.

Response: Figure 3 (now Figure 4) was moved to the end of Results section

8. Materials and methods

- Ethical approval. Your research was from 2004 year, but Board approval was dated Sept. 2013. How could You explain this difference?

Response: We have searched our files retrospectively from 2004 to 2019; the IRB application was submitted and approved in 2019.

- Please, describe the zone and methods of biopsy/material taking.

Response: The following was added as suggested:

"In general, all the oral lesions were biopsied by scalpel with patients under local an-esthesia. Those defined, exophytic, pedunculated or broad-based lesions were usually re-moved per excisional biopsy. Flat, extensive and usually less well-defined lesions were sampled as an incisional biopsy procedure. Removed tissues were fixed in 10% formalin solution for 24 h and then processed by an automatic benchtop tissue processor (Leica TP1020, Leica Biosystems, Deer Park, IL, USA). Four-micron width sections were prepared and then routinely stained with hematoxylin and eosin (H&E)".

- Statistics. It is too small description, please, write this section with details. Also, you have many comparisons that require a correction. Please, write results according to Bonferroni of FDR correction.

Response: The entire paragraph on the statistical analysis has been revised, with detailed and specific reference to tests that were used (lines 554-561)

- Section ‘Conclusion’ is missed. Please, write it.

Response: Conclusions were written as a separate paragraph and further refined in the revised version (lines 447-453)

"In conclusion, there are a few cases with microscopic evidence of CH embedded into oral epithelium without typical signs/symptoms of OC, including healthy/non-smokers individuals. This finding could be referred to as "hidden" OC. It can be assumed that the oral mucosae harbor focal niches, where local mechanical trauma can facilitate the morphological transformation of candida into a potentially pathogenic state, leading to a focal occult OC. More concrete evidence for the mechanism driving the mechanical force-related, pathological transformation of candida in the oral mucosae, is needed."

Round 2

Reviewer 1 Report

I accept in this form.

Author Response

We thank the reviewer for accepting our revised version. 

Reviewer 2 Report

Dear Authors, 

Thank You for Your great work.

Although, several parts must be corrected.

Abstract

I meant You need to add names of sections, exactly words ‘Background’, ‘Aim’, ‘Materials and methods’, ‘Results’, ‘Conclusion’.

Table 1. Please, add p-value. It is exactly ‘probability’ not the OR written by

You. Also, it is the same for other table. 

Figure 3 still remains in ‘Discussion’. Please, remove it.

Statistics. Please, write the method for distribution normality checking and add in tables columns for p-value after Bonferroni correction results besides the primary p-values.

Sincerely, Reviewer

Author Response

We thank the reviewer for the additional comments.

Our point-by-point responses follow. The changes done in this second round were marked in yellow. We hope that the current revised version of the manuscript will now be appropriate for publication.

Sincerely, The authors

  1. Abstract: I meant You need to add names of sections, exactly words 'Background', 'Aim', 'Material and methods', 'Results', 'Conclusion'.

Response: These headings were added in the abstract, p.1, lines 17-29

  1. Table 1: Please add p-value. It is exactly 'probability' not the OR written by You. Also, it is the same for the other table

Response: In Table 1 only descriptive statistics was done, as the number of patients was small; the original Table 1, without 'probabilities', has been returned. Table 2 does show p-values

  1. Figure 3 still remains in 'Discussion'. Please remove it.

Response: Figure 3 (now Figure 4) was removed to end of 'Results' already in the first round of corrections, (reference in the text to Figure 4 on p. 6 lines 271-272 and Figure 4 on p. 7, lines 295-311)

  1. Please, write the method for distribution normality checking and add in tables columns for p-value after Bonferroni correction results besides the primary p-values.

Response: Method for distribution of normality was added in 'Material and Methods' – p. 10, lines 571-2.

We have added the p-values after Bonferroni correction in Table 2 as a footnote because all those primary p-values that were <0.001 remained so after the correction and only one p-value has been modified after the corrections (p. 4, lines 142-3). Table 1, as mentioned above, showed only descriptive statistics. P-value after Bonferroni was added also on p. 5, lines 173-4.